# Automatic Bone Segmentation from MRI for Real-Time Knee Tracking in Fluoroscopic Imaging

**DOI:** 10.3390/diagnostics12092228

**Published:** 2022-09-15

**Authors:** Brenden Robert, Pierre Boulanger

**Affiliations:** Department of Computing Science, University of Alberta, Edmonton, AB T6G 2E8, Canada

**Keywords:** bone segmentation, neural networks, patellofemoral syndrome, MRI, CT

## Abstract

Recent progress in real-time tracking of knee bone structures from fluoroscopic imaging using CT templates has opened the door to studying knee kinematics to improve our understanding of patellofemoral syndrome. The problem with CT imaging is that it exposes patients to extra ionising radiation, which adds to fluoroscopic imaging. This can be solved by segmenting bone templates from MRI instead of CT by using a deep neural network architecture called 2.5D U-Net. To train the network, we used the SKI10 database from the MICCAI challenge; it contains 100 knee MRIs with their corresponding annotated femur and tibia bones as the ground truth. Since patella tracking is essential in our application, the SKI10 database was augmented with a new label named *UofA Patella*. Using 70 MRIs from the database, a 2.5D U-Net was trained successfully after 75 epochs with an excellent final Dice score of 98%, which compared favourably with the best state-of-the-art algorithms. A test set of 30 MRIs were segmented using the trained 2.5D U-Net and then converted into 3D mesh templates by using a marching cube algorithm. The resulting 3D mesh templates were compared to the 3D mesh model extracted from the corresponding labelled data from the augmented SKI10. Even though the final Dice score (98%) compared well with the state-of-the-art algorithms, we initially found that the Euclidean distance between the segmented MRI and SKI10 meshes was over 6 mm in many regions, which is unacceptable for our application. By optimising many of the hyper-parameters of the 2.5D U-Net, we were able to find that, by changing the threshold used in the last layer of the network, one can significantly improve the average accuracy to 0.2 mm with a variance of 0.065 mm for most of the MRI mesh templates. These results illustrate that the Dice score is not always a good predictor of the geometric accuracy of segmentation and that fine-tuning hyper-parameters is critical for improving geometric accuracy.

## 1. Introduction

Patellofemoral pain (PFP) is the most frequent cause of knee pain. It frequently occurs in teenagers, manual labourers, and athletes. Patellofemoral pain syndrome may be caused by overuse, injury, excess weight, a kneecap that is not aligned correctly (patellar tracking disorder), or changes under the kneecap. Many researchers have shown that PFP is a precursor to early osteoarthritis [1,2], the leading cause of disability in the North American elderly population. Despite considerable efforts to improve the effectiveness of treatment interventions, 70% of individuals with PFP had persistent or recurrent symptoms 5–20 years after the intervention. Current interventions’ poor long-term outcomes highlight the necessity of better understanding the causes of PFP and its contributing factors. Generally speaking, patellar misalignment resulting in abnormal stress distribution is considered the primary mechanism for PFP [3,4]. However, the exact cause is unknown. This is due to the inability to evaluate in vivo three-dimensional (3D) patellofemoral joint (PFJ) motion during dynamic activities. Motion tracking in clinical studies is commonly based on surface marker video motion analysis, which suffers from measurement inaccuracy produced by soft tissue artefacts [5]. Therefore, researchers are investigating PFJ kinematics using 3D imaging modalities instead. In a paper by Esfandiarpour et al. [6], a new technique for real-time tracking of the knee bone structure using fluoroscopic imaging was proposed and tested. The methodology combined a computed tomography (CT) scan and bi-planar fluoroscopy techniques (see Figure 1a) to examine the knee motion with six degrees of freedom during the squatting of individuals with the syndrome and without. The calibration target shown in Figure 1b was used to correct the geometric distortion of the fluoroscopic images and determine the two imagers’ relative projection geometries. The calibration target was placed close to each image intensifier. A calibration algorithm compensated for radial distortions by using a matrix of small stainless steel spheres located at known positions, as shown in Figure 1c. Each image intensifier was independently corrected by using a radial distortion model. Then, an orientation plate (Figure 1d) was placed in the shared image acquisition area and imaged with two fluoroscopic sensors. One can compute the imaging sensors’ relative positions and orientations using the two images. Tracking the relative positions and orientations of the bones in a knee (tibia, femur, and patella) over time requires that a set of 3D mesh templates (one for each bone) must be extracted from CT imaging and individually segmented by using a simple thresholding technique (see Figure 2). One must track the projected image into the two by-plane fluoroscopic sensors to compute each knee bone’s position and orientation relative to the X-ray source (see Figure 3). Knowing the projective transform between the X-ray source and the two sensors from calibration, one can then infer what position and orientation each bone must have to produce these two images. The fluoroscopic imaging system captured two X-ray videos of the knee bones during motion from orthogonal viewpoints; one can see the resulting video sequence in Figure 4. Using an iterative re-projection software from Innomotion, Inc. (Shanghai, China) (http://www.innomotion.biz/ (accessed on 20 June 2022)) called FluoMotion, one can infer the three bones’ positions and orientations from the two orthogonal fluoroscopic image sequences. In Figure 5, one can see the projection of the knee bones for one frame, and in Figure 6, one can see a block diagram of the 3D-to-2D registration algorithm used, which is very similar to the algorithm described in [7].

The problem with creating knee models from CT imaging is that it exposes patients to extra ionising radiation, which adds to fluoroscopic imaging. One possible solution is to generate the templates from MR imaging (MRI) instead. Indeed, many automatic methods have been proposed to create a knee model [8,9], but none can be as precise and robust as manual or semi-automatic methods. Furthermore, MRIs are often very noisy, and the contrast of the bones with their surroundings is frequently poor and dependent on the imaging protocol. This paper presents the application of U-Net to the segmentation of MR images to extract those bone templates automatically at an average accuracy of 0.2 mm for use in fluoroscopic tracking applications. The accuracy of the 3D template was set based on the results described in [6] and the limited resolution of MR images.

### 1.1. Contributions of the Paper

The first contribution of this paper is to demonstrate that, despite popular belief, the Dice score is not always a good predictor of the geometric accuracy of segmentation. The second contribution is that this accuracy can be significantly improved by changing network hyper-parameters, such as the threshold used in the network’s last layer. Finally, accurate 3D knee bone templates (better than 0.5 mm) can be created without having to expose patients to the extra ionising radiation produced by a CT imaging system.

### 1.2. Organization of the Paper

In Section 2, we will review the current state of the art of bone segmentation from MRI. Section 3 will describe the 2.5D U-Net architecture used for segmenting the MRIs. We used the SKI10 database from the MICCAI challenge, which contains 100 knee MR images with the corresponding annotated femur and tibia segmentations produced by medical experts to train the network parameters. Section 4 will describe the training process and the geometric comparison of the U-Net segmentation results to the SKI10 label maps, and it will present a geometric validation of the mesh using a gold standard. We will then conclude in Section 5 by discussing the advantages of the proposed method.

## 2. Related Work

As discussed previously, the accurate segmentation of knee bones from MRI is imperative for quantitatively assessing patellofemoral syndrome. What is essential is to create 3D knee bone templates that can be used to track knee motion in the fluoroscopic image sequence. Current clinical practice requires a radiologist to perform manual segmentation slice by slice by using manual or semi-automated tools [10]. The process is costly in terms of time and money and can easily take 3–4 h for a single scan. Repeatability and accuracy are also an issue, as this process is observer-dependent [11]. Numerous papers have evaluated segmentation accuracy by using public-domain MRI datasets, such as MICCAI SKI10 accessed (https://ski10.grand-challenge.org/ (accessed on 20 June 2022)) and OAI Imorphics (http://imorphics.com/ (accessed on 20 June 2022)), where bone and cartilage boundaries have been manually segmented by radiologists, and these are used as a gold standard to train and compare the results of automated algorithms. The most common metric for comparing the segmentations produced by automated algorithms with the gold standard is the Dice coefficient (DSC), which is defined as:(1)$$\mathrm{DSC}\left(S,R\right)=\frac{2|S⋂R|}{|S|+|R|}\times 100\%$$
where *S* is the bone’s surface segmented by the network and *R* is the ground truth. The coefficient is 100% when the segmentation is perfect and 0% when it is false. The Dice coefficient is more adapted to the segmentation of small objects because it is independent of their size in the image instead of the pixel accuracy function, which depends on the proportions between the foreground and the background. The other metrics used to characterise the segmentation results are geometric, and they include the following:The average surface distance between the boundary ∂S and ∂R for one MRI *n* is defined as:
(2)$${\hat{S}}_{n}=\frac{1}{N_{s}}\sum_{i=1}^{N_{s}}\underset{y\in \partial R}{\inf}∥s_{i}-y∥$$The root mean square surface distance for one MRI *n* is defined as:
(3)$${\hat{R}}_{n}=\sqrt{\frac{1}{N_{s}}\sum_{i=1}^{N_{s}}\inf_{y\in \partial R}{∥s_{i}-y∥}^{2}}$$
where $N_{s}$ is the number of 3D points located on the segmented bone surface boundary ∂S. The calculation of the distance between 3D points $s_{i}$ and y∈∂R corresponds to the minimum distance between a 3D point $s_{i}$ and a point y located on reference surface boundary ∂R. Statistics are represented as ASD = $avr_{M}\left({\hat{S}}_{n}\right)\pm var_{M}\left({\hat{S}}_{n}\right)$ and RSD = $avr_{M}\left({\hat{R}}_{n}\right)\pm var_{M}\left({\hat{R}}_{n}\right)$, where $avr_{M}\left(*\right)$ is the average value and $var_{M}\left(*\right)$ is the variance for a group of *M* test MRIs.

Other geometric metrics can be used to characterise the segmentation results of cartilages, such as volume difference (VD) and volume overlap error (VOE), which, in our case, is not necessary.

Current segmentation techniques can be classified into two categories: semi-automatic and automatic. Semi-automatic methods include thresholding [12], region growing [13], deformable models [14,15], and clustering [16]. Most of these approaches reduce the time for segmenting bone and cartilage in MRI, but are still heavily manual and time-consuming. Fully automated algorithms include k-nearest neighbour voxel classification by [17], which is used mainly for cartilage segmentation. In [18], one can find an excellent review of automated bone and cartilage segmentation algorithms. Tamez et al. [19] and Shan et al. [20] used approaches based on multi-atlas approaches. In these approaches, human experts segmented MRI series that were used as reference atlases for a multi-atlas segmentation algorithm. The methodology created good knee segmentation that was then used to extract articular cartilage volume, surface area, thickness, and subchondral bone plate curvature. Typically, these algorithms’ DSC values are 88% for the tibia bone and 89% for the femur bone. The RSD values are 1.4 and 1.0 mm for the femur and tibia, respectively. Liu et al. [21] used deep convolutional neural networks and a 3D deformable approach to segment bones and cartilages. The segmentation results trained by the SKI10 database for the femur bone were: DSC = 64.1%, ASD = 0.56±0.12, and RSD = 1.08±0.21. For the tibia bone, the results were DSC = 64.1%, ASD = 0.50±0.14, and RSD = 1.09±0.28. Numerous deep learning segmentation algorithms found in the recent literature were compared by Ambellan et al. [22] with their algorithm based on statistical shape knowledge and convolutional neural networks. For the femur bone, the results were DSC = 98%, ASD = 0.43±0.13, and RSD = 0.74±0.27. For the tibia bone, the results were DSC = 98.5%, ASD = 0.35±0.07, and RSD = 0.59±0.19. Almajalid et al. [23] used an approach similar to the one proposed in this paper. The DSC values for the tibia were 96.83%, and those for the femur were 97.92%. Unfortunately, no geometric distance metrics were provided. Chen et al. [24] proposed a bone and cartilage segmentation algorithm based on a 3D DNN using adversarial loss for a prior shape constraint. The DSC scores were 98% for the tibia and femur bones. For the tibia, ASD = 0.38±0.15 and RSD = 0.69±0.37. For the femur, ASD = 0.29±0.07 and RSD = 0.52±0.12.

## 3. Method

### 3.1. SKI10 Training Datasets

This paper used a 2.5D U-Net architecture to segment knee bones using the SKI10 database from the SKI10 MICCAI challenge containing 100 knee MRIs corresponding to professionally annotated femur and tibia segmentation images. The SKI10 database is from MRI images originating from Biomet, Inc. The data were acquired at over 80 different centres in the USA using machines from all major vendors, i.e., General Electric, Siemens, Philips, Toshiba, and Hitachi. All images were acquired in the sagittal plane with a rectangular pixel spacing of 0.4 mm and a slice distance of 1 mm. No contrast agents were used. The field strength was 1.5 Tesla in about 90% of the cases. The rest was acquired at 3 Tesla, with some images acquired at 1 Tesla. The MRI sequences show a considerable variety: Most images used T1-weighting, but some were also obtained using T2-weighting. In addition, many images used gradient echo or spoiled gradient echo sequences, and fat suppression techniques were also standard. All MRI images were segmented interactively by experts at Biomet, Inc. into four categories corresponding to the femur, femoral cartilage, tibia, and tibial cartilage. Unfortunately, the SKI10 database did not provide a patella category. To remediate this situation, we asked one of the radiologists at the University of Alberta to add patella labels with the name *UofA Patella* to the database.

### 3.2. Segmenting Bone MRI Using 2.5D U-Net

The first use of a deep neural network for medical image segmentation was proposed by Ciresan et al. [25] for the segmentation of electron microscopy images. This was a fully connected approach where a classifier was applied to each slice of pixels using a sliding window around the pixel. A drawback of this approach is that there are too many overlaps and redundant computations to be efficient. Ronneberger et al. [26] redesigned this fully connected network by using convolutions instead and got better results by proposing a new architecture called U-Net. U-Net is based on the notion of an encoder–decoder network in which the input images are inputted into an encoder network to extract high-level features. These features are then sent to a decoder network to restore the spatial information to classify the pixels. The U-Net architecture has produced excellent results for many medical image segmentation applications. The U-Net architecture was first proposed to deal with 2D images, but since MRIs are volume datasets, one has two possibilities to segment them. First, one can use a generalization of U-Net called 3D U-Net [27] that can deal with native volume data segmentation. Second, one can segment each slice independently by using a 2D U-Net and can then combine the labelled results to generate a 3D mesh model. These 2.5D approaches described in [28] are inspired by the fact that 2.5D has richer spatial information of neighbouring pixels with lower computational and memory costs than those of the 3D U-Net architecture. In this paper, we implemented the 2.5D U-Net approach for the following reasons:Ease of implementation on a GPU with limited global memory;Significant reduction in training and prediction time;Leveraging of the higher spatial resolution in the axial direction.

In addition to training, the network weights when using back-propagation and other networks’ hyper-parameters must also be optimised. Hyper-parameters are the variables that determine the network structure and the variables that determine how the network is trained. They include:Batch size: The mini-batch size is the number of sub-samples given to the network, after which a parameter update happens.Learning rate: The learning rate defines how quickly a network updates its parameters.Momentum: Momentum helps to know the direction of the next step with the knowledge of the previous steps.Loss/cost function: The loss function measures how wrong the model is.Activation function: Activation functions are used to introduce nonlinearity to models, which allows deep learning models to learn nonlinear prediction boundaries.Optimiser: An optimization algorithm is used.Architecture: This indicates the number of hidden layers and units.Dropout: Dropout is a regularization technique for avoiding overfitting (increasing the validation accuracy), thus increasing the generalization power.Initialiser: This is the network weight initialization.Validation split: This is the ratio of validation data.Test split: This is the ratio of test data.Epoch number: The number of epochs is the number of times the whole set of training data is shown to the network while training.

One can see the network architecture in Figure 7 and a description of our implementation using the TensorFlow framework in Figure 8.

### 3.3. Geometric Validation of the Segmentation

The problem with the Dice score is that one does not measure the geometric differences between a 3D mesh produced by 2.5D U-Net and its SKI10 ground truth for a score of 98% instead of 90%. This is critical for our application, as we will use the geometry extracted from the segmentation as a 3D template to be back-projected into the fluoroscopic image sequences. Therefore, the geometric difference between the 3D mesh produced by the 2.5D U-Net segmentation and the SKI10 ground truth will be computed as follows:Generate a 3D mesh template of one of the knee bone structures using a marching cube algorithm from the SKI10 ground truth;Generate a 3D mesh template of one of the knee bone structures using a marching cube algorithm from the segmentation produced by 2.5D U-Net;Register the MRI and SKI10 3D mesh templates for each patient using a robust ICP algorithm [29];Compute the Euclidean distance between the MRI and SKI10 3D mesh templates;Display colour-coded differences in the 3D rendering of the mesh geometry.

### 3.4. Pre-processing the Training Datasets

The 100 SKI10 MRIs were randomly divided into a training set of 70 MRI volumes with a corresponding label map and a test set of 30 MRI volumes. Since the segmentation algorithm used a 2.5D U-Net, it was necessary to re-sample the volume dataset to a resolution of 128 × 128 pixels to deal with GPU memory limitations. We used a bi-cubic interpolation algorithm provided by the ITK library. To train a 2.5D U-Net network, one must give all training set input volumes and their corresponding labelled map. The total combined number of slices for the training set was 7000. This conversion was a complex task because the SKI10 data were far from being normalised to images of 128 × 128 pixels. One can see the pre-processing pipeline in Figure 9. All volumes were of different sizes, and volumes containing the labels were not binary because they included bone and cartilage types. To process, each MRI training volume dataset must be combined into a large array of sizes (*S* × *m* × *m* × 2 bytes) that combine all training volume MRI values with their corresponding segmentation values. *S* is the total number of axial slices in the training set equal to 7000. The variable *m* is the new resolution of each axial slice and is equal to 128, where two values are attributed to each element of this large array. The first value is the re-normalised value of each voxel between 0 and 255 based on the min–max of all volumes in the training dataset. The second is the corresponding voxel classification as bone or cartilage, and it is defined as the gold standard. Once the data were combined and pre-processed, the network was ready to be trained using a back-propagation algorithm provided by the TensorFlow (https://www.tensorflow.org/ (accessed on 20 June 2022)) software.

## 4. Experimental Results

First, pre-processing was applied to the 70 training set volumes with corresponding label maps for the tibia and femur. During the training with cartilage, the labels were discarded. Then, the TensorFlow back-projection algorithm set the weight using a momentum acceleration technique. All biases were initialised to 0, and the weight values were initialised to random numbers. The network reached a plateau after about 70 epochs. In general, stopping training at this point was better to avoid overfitting. One can see the evolution of the Dice score relative to the number of epochs for the training set (blue curve) in Figure 10. The Dice coefficient on the graph was calculated from an average of thousands of images from the training set volumes. If one observes the segmentation results volume by volume, important variations can be observed depending on the MRI quality and the accuracy of the label map. By filtering the label map with a morphological opening operator during pre-processing, we eliminated the labelled one-pixel regions that interfered with the training process. With the filter, the training Dice coefficient converged to an excellent value of 98%, which compared favourably with two of the best algorithms reported in [23,24]. However, on more complicated volumes, the coefficient could fall to 90%. The validation set was composed of 30 MRI volumes. The data allowed us to judge the segmentation quality at regular intervals without interfering with the training process. One can see the evolution of the global Dice score (orange curve) relative to the number of epochs in Figure 10. This Dice score closely followed the evolution of the training Dice score, but converged instead to a value of 95%.

The training phase took 20 h on a GeForce 980 TI GPU and 5 h on a Geforce 1080 TI GPU. A 3D mesh was generated using a marching cube algorithm following training. In Figure 11a, one can see the mesh produced for one instance of the femur 3D mesh created by the 2.5D U-Net, and in Figure 11b, one can see its corresponding mesh from the SKI10 label map. Likewise, one can see the 3D mesh produced by U-Net in Figure 12a and the corresponding 3D mesh from the SKI-10 label map for the tibia in Figure 12b. In Figure 13a, one can see the mesh produced for one instance of the patella 3D mesh created by the 2.5D U-Net, and in Figure 13b, one can see its corresponding mesh from the UofA label map.

In Figure 14, Figure 15 and Figure 16, one can see the signed difference map between the U-Net mesh and the labelled meshes of the femur, tibia, and patella, respectively. A colour-coded scale displays the differences in mesh geometry in both images. As one can see, even though the ASD value was 1.5±0.58 mm and RSD = 2.5±0.8 mm for the tibia and femur, there were regions where the differences exceeded 6.0 mm, which was unacceptable for our application. For the patella, the ASD value was 0.56±0.2 mm and RSD = 1.2±0.54 mm. These significant differences were due to areas where the MRI had more diffuse boundaries.

### Improving Geometric Accuracy by Optimising Hyper-Parameters

Fortunately, by changing the classification threshold in the final layer of the U-Net from the standard 0.5 to an optimal value of 0.42, we were able to reduce the ASD value to 0.2±0.065 mm and RSD = 0.5±0.14 for all tibia and femur models. For the patella, the ASD value was 0.15±0.07 mm and RSD = 0.6±0.13 mm. As a result, in Figure 17, Figure 18 and Figure 19, one can see the new signed distance map for the femur, tibia, and patella, respectively. Therefore, even though the Dice score was very high (98%), a geometric comparison was necessary in order to assess the proper segmentation accuracy and help optimise the network’s hyper-parameters.

## 5. Conclusions

In this paper, we showed that it is possible to extract accurate enough (≤0.5 mm) 3D bone templates that can be used to track knee motion using a fluoroscopic imaging system. Our study found that, for most segmented bone templates (tibia, femur, and patella), the DSC values were 98%. Additionally, ASD = 0.2±0.065 mm and RSD = 0.5±0.14; these values for the tibia, femur, and patella were better than those obtained with most algorithms. In fact, our approach is superior to the well-known SegNet algorithm [21] (DSC = 64% with ASD = 0.56 mm) and compares well with two of the best segmentation algorithms [23,24] (DSC = 98.5%, ASD = 0.4 mm) found in the literature. As stated in [23], the 2.5D U-Net is more cost-effective in terms of memory needs and calculation time. This paper also shows that a good Dice score is not always a good indicator of geometric precision, and it can be improved by optimising the network’s hyper-parameters. Even though the original SKI10 database did not include patella labels, by adding extra manual segmentation for the patella from one of our medical experts, we were able to get good segmentation results. This is very encouraging, as this work shows that MRI scans can be used instead of CT to create accurate bone templates and significantly reduce exposure to ionising radiation.

## Figures and Tables

**Figure 1 diagnostics-12-02228-f001:**
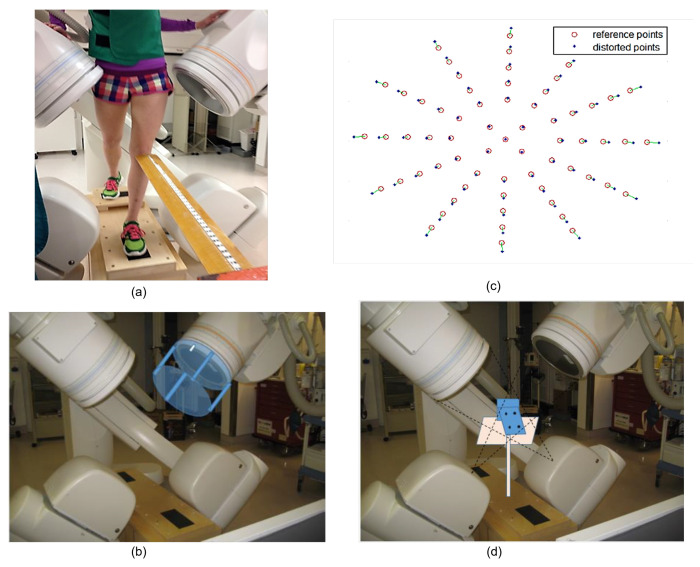
Experimental set-up for fluoroscopic imaging of the knee. (**a**) Bi-plane imaging system. (**b**) Calibration jig made of small stainless steel spheres located at known positions. (**c**) Calibration image. (**d**) An orientation plate was used to determine the common coordinate system between the two imagers.

**Figure 2 diagnostics-12-02228-f002:**
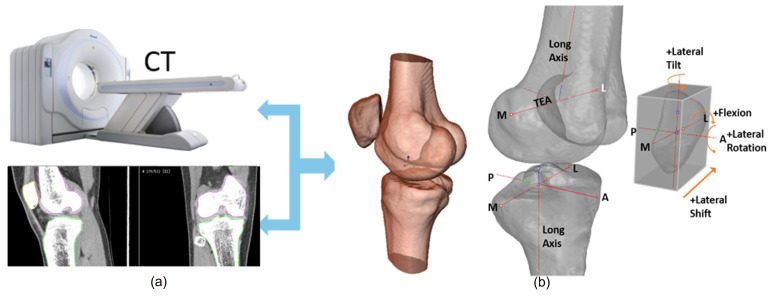
(**a**) Creation of 3D knee bone templates from a series of CT scans captured in the knee extension position. (**b**) Definition of the coordinate systems of the femur, the tibia, and the patella used to quantify the patellar tracking parameters and tibiofemoral joint angle: M, medial; L, lateral; A, anterior; P, posterior; TEA, trans-epicondylar axis.

**Figure 3 diagnostics-12-02228-f003:**
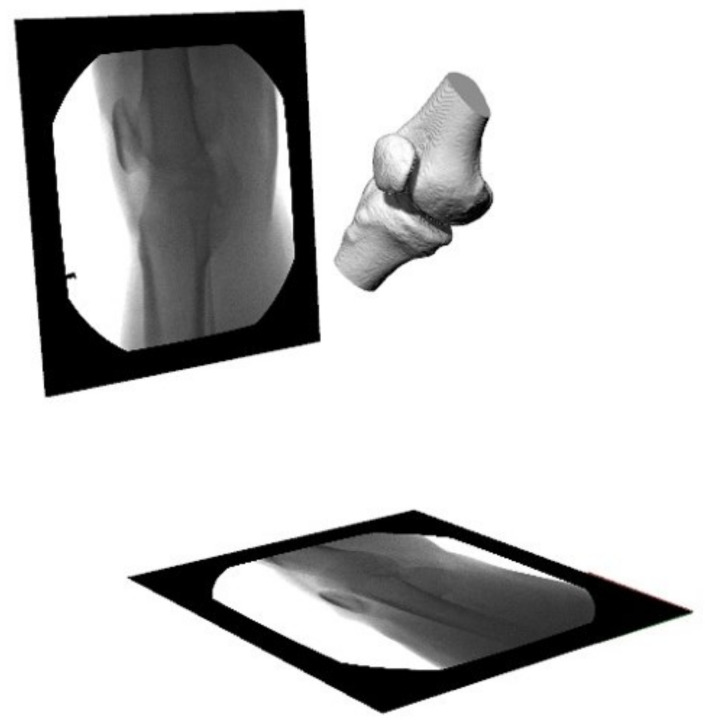
The virtual dual-orthogonal fluoroscopic system duplicates patellofemoral joint motion using a 3D-to-2D registration technique.

**Figure 4 diagnostics-12-02228-f004:**
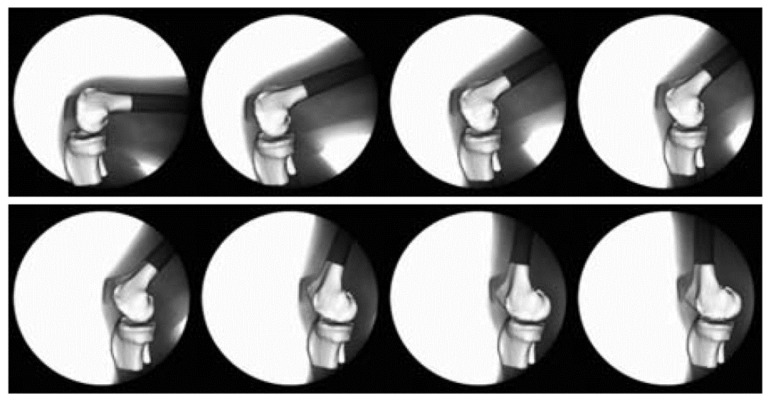
Example of 3D knee templates re-projected into the fluoroscopic image sequence.

**Figure 5 diagnostics-12-02228-f005:**
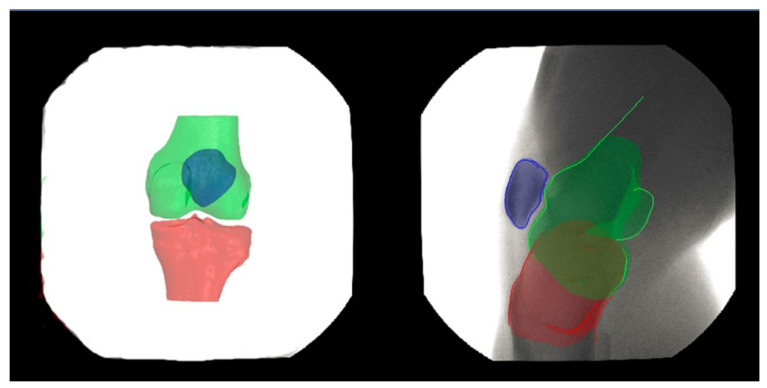
Example of a 3D knee template re-projected into one fluoroscopic image. The red color correspond to the tibia, the green color to the femur, and the blue color to the patella.

**Figure 6 diagnostics-12-02228-f006:**
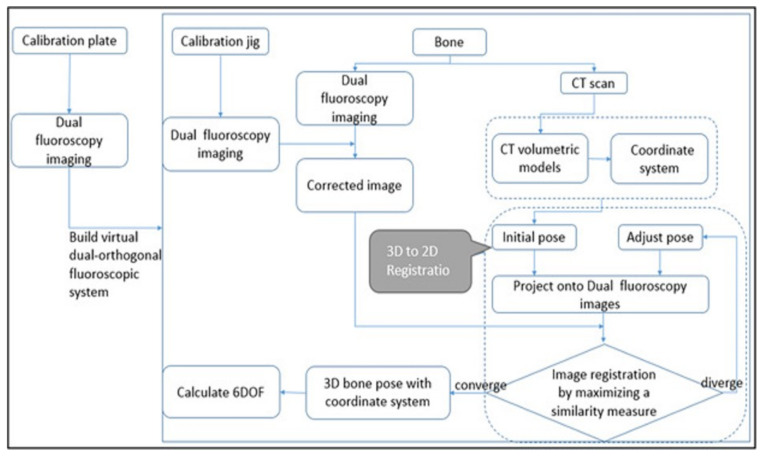
Block diagram of the 3D-to-2D re-projection algorithm.

**Figure 7 diagnostics-12-02228-f007:**
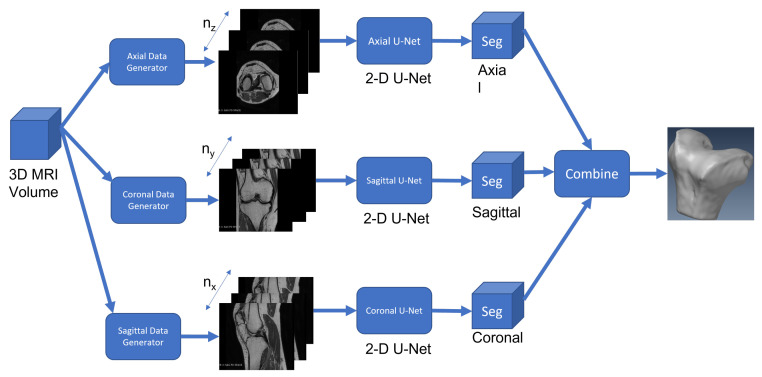
The 2.5D U-Net architecture.

**Figure 8 diagnostics-12-02228-f008:**
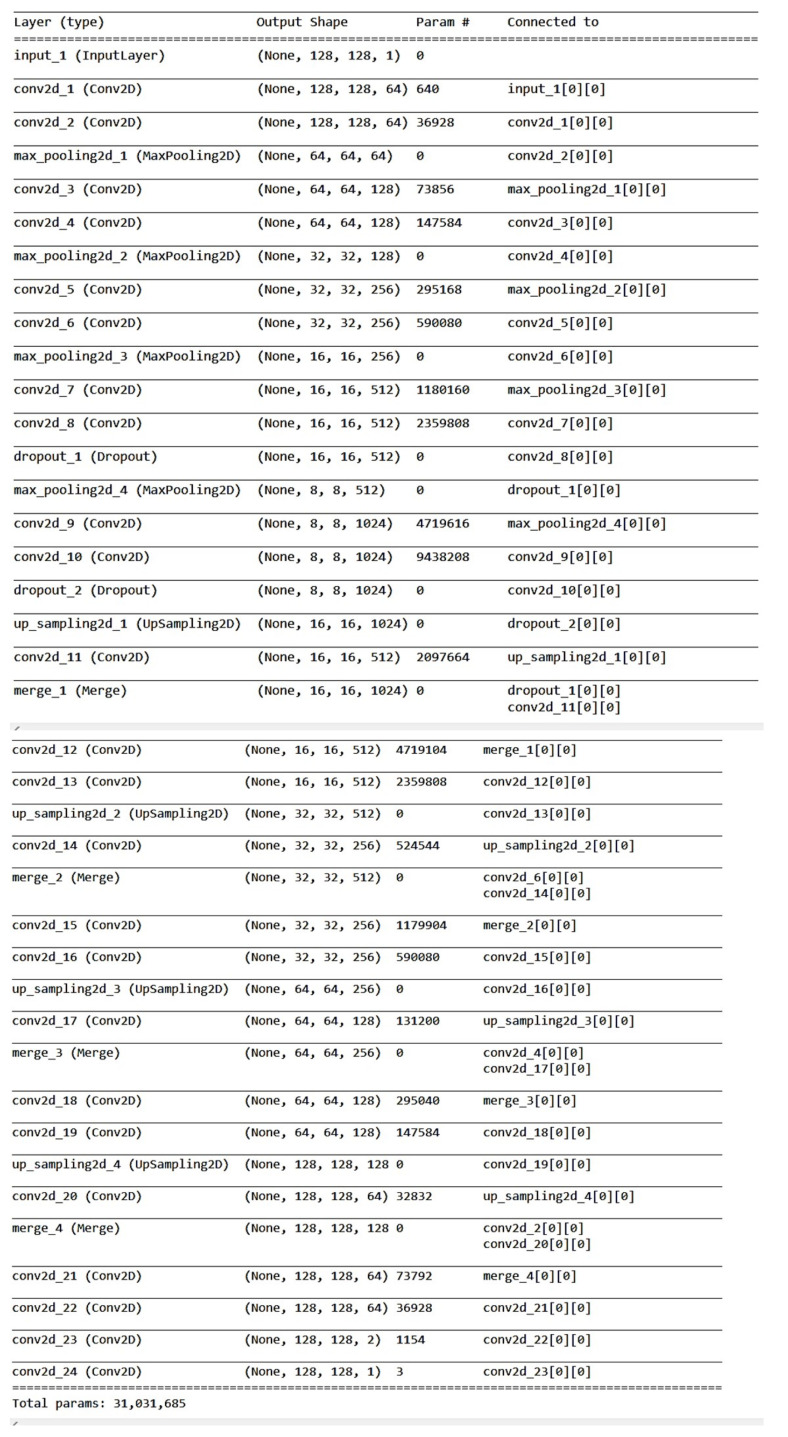
TensorFlow definition of the 2D U-Net used three times in the 2.5D version.

**Figure 9 diagnostics-12-02228-f009:**
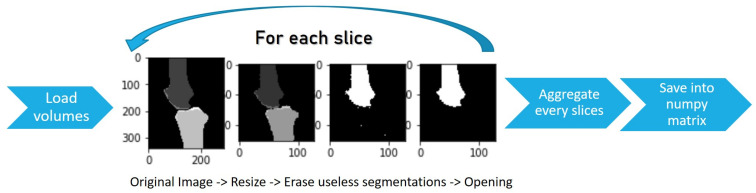
Segmentation pre-processing pipeline.

**Figure 10 diagnostics-12-02228-f010:**
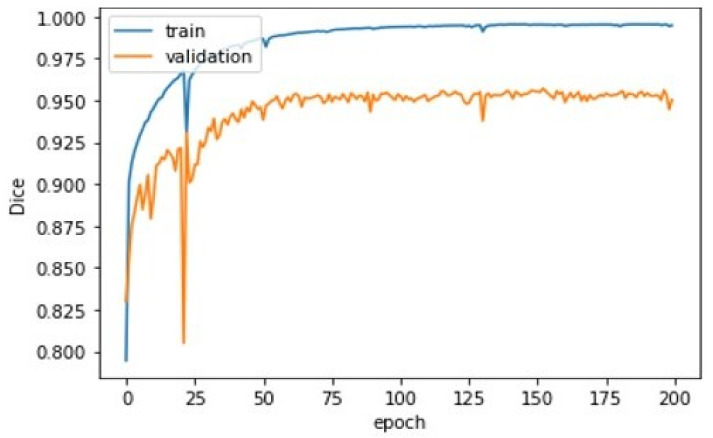
(Blue) Evolution of the Dice score for 70 volumes from the SKI10 database and (Orange) evolution of the Dice score for the validation set relative to the number of epochs.

**Figure 11 diagnostics-12-02228-f011:**
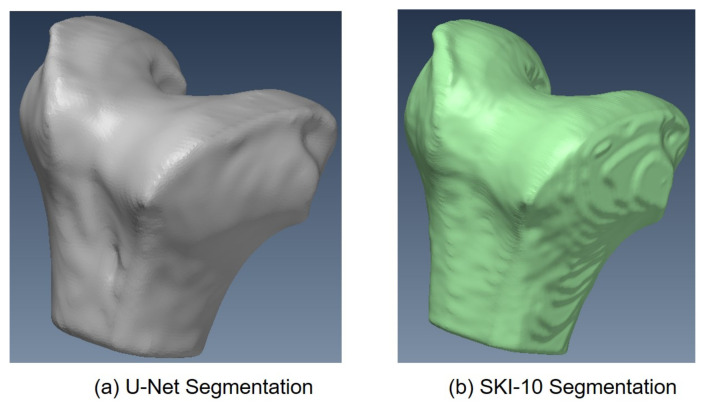
Femur 3D mesh model produced by (**a**) U-net and (**b**) the SKI10 ground truth.

**Figure 12 diagnostics-12-02228-f012:**
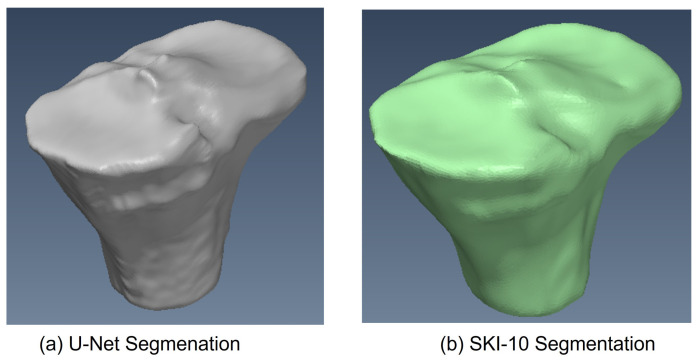
Tibia 3D mesh model produced by (**a**) U-Net and (**b**) the SKI10 ground truth.

**Figure 13 diagnostics-12-02228-f013:**
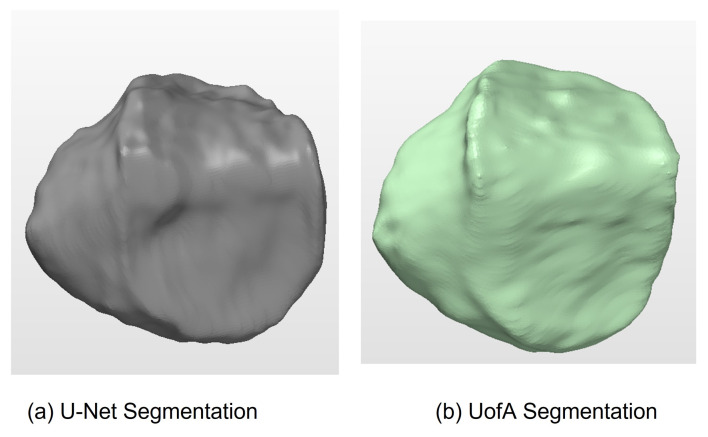
Patella 3D mesh model produced by (**a**) U-Net and (**b**) the UofA ground truth.

**Figure 14 diagnostics-12-02228-f014:**
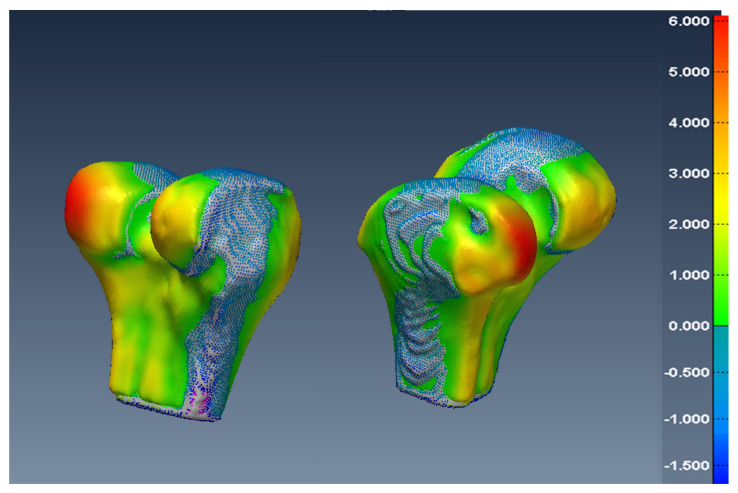
Two views of the colour-coded signed distance in mm between the U-Net 3D mesh and the SKI10 3D mesh for the femur.

**Figure 15 diagnostics-12-02228-f015:**
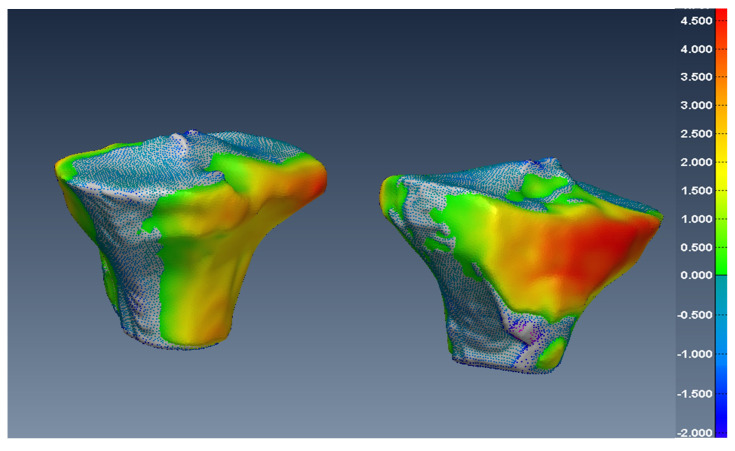
Two views of the colour-coded signed distance in mm between the U-Net 3D mesh and the SKI10 3D mesh for the tibia.

**Figure 16 diagnostics-12-02228-f016:**
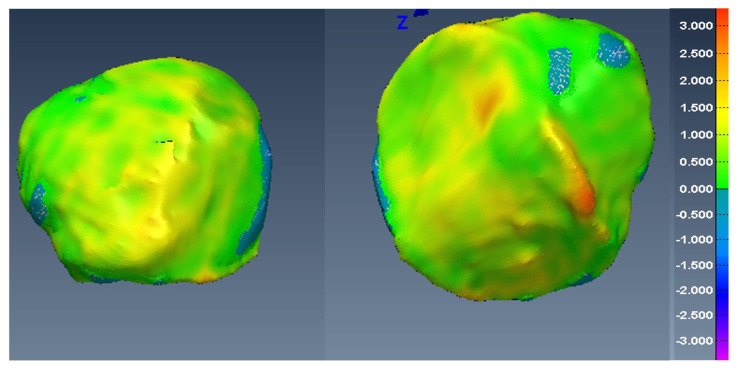
Two views of the colour-coded signed distance in mm between the U-Net 3D mesh and the UofA 3D mesh for the patella.

**Figure 17 diagnostics-12-02228-f017:**
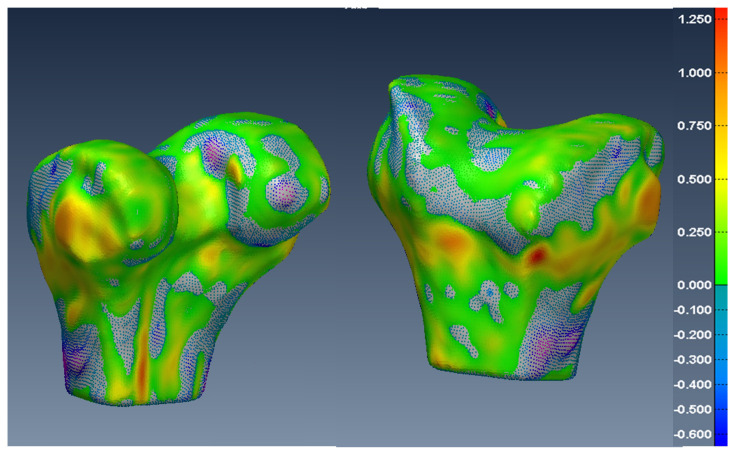
Two views of the colour-coded signed distance in mm between the U-Net mesh and the SKI10 mesh for the femur with the optimal threshold.

**Figure 18 diagnostics-12-02228-f018:**
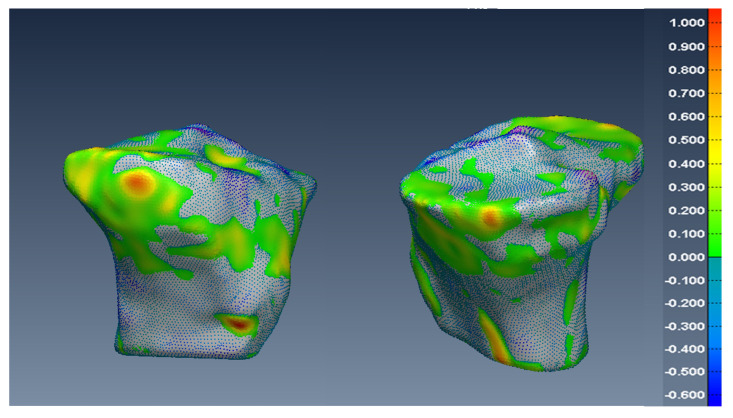
Two views of the colour-coded signed distance in mm between the U-Net mesh and the SKI10 mesh for the tibia with the optimal threshold.

**Figure 19 diagnostics-12-02228-f019:**
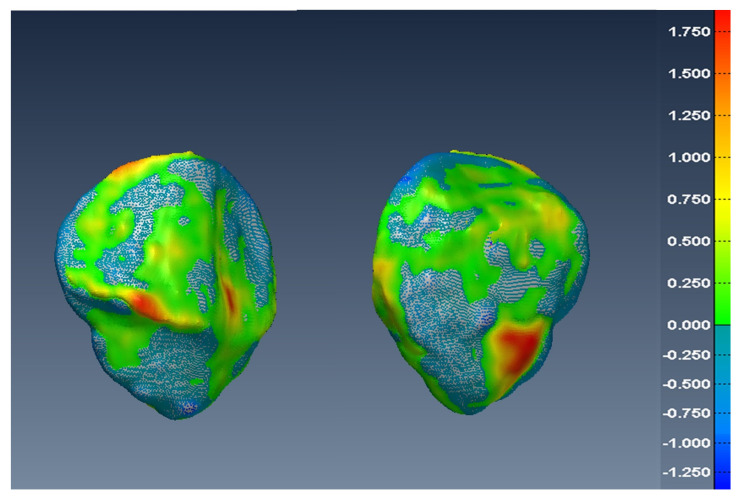
Two views of the colour-coded signed distance in mm between the U-Net mesh and the UofA mesh for the patella with the optimal threshold.

## Data Availability

The data used to trained the network is available at MICCAI SKI10 database located a https://ski10.grand-challenge.org/ (accessed on 20 June 2022).

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
