# Peer review of "Automatic Bone Segmentation from MRI for Real-Time Knee Tracking in Fluoroscopic Imaging"

_diagnostics, 2022, doi:10.3390/diagnostics12092228_

Round 1
Reviewer 1 Report
Authors may revise the abstract to elaborate more on the problem statement and their findings and contributions.
Authors may elaborate more on the novelty of their work. How it contributes to the literature.
Thorough proofreading is recommended.
A few of the references are missing some information, so may complete them critically.
A few of the figures are readable but can be improved in their quality.
The conclusion is not clear and needs revision and clarity and alignment with the abstract and title.
Provided references are better enough. However, authors are recommended to consider more latest and related.
Author Response
Authors may revise the abstract to elaborate more on the problem statement and their findings and contributions.
We have changed the abstract to define better the paper’s objectives and its results. The original contribution of the paper is to show that despite popular belief, the Dice score is not necessarily a good predictor of the geometric accuracy of segmentation results and that geometric accuracy can be improved by changing some of the network’s hyper-parameters such as the threshold at the last layer on the network.
Authors may elaborate more on the novelty of their work. How it contributes to the literature.
An additional sub-section was added in the introduction describing the main contributions of the paper.
Thorough proofreading is recommended.
A proofread of the paper was performed by a professional English-speaking editor.
Some references are missing some information, so may complete them critically.
The literature review was redone to describe better recent algorithms and their results.
A few of the figures are readable but can be improved in their quality.
Figures 8 and 9 were enlarged to improve visibility. Color code for the geometric comparison was enlarged to be more readable.
The conclusion is not clear and needs revision and clarity, and alignment with the abstract and title.
The conclusion text was modified to highlight better the results obtained.
Provided references are better enough. However, authors are recommended to consider more latest and related.
We have added an extra 2022 paper in addition to the paper by Almajalid (2022)
Chen Hao, Zhao Na, Tan Tao, Kang Yan, Sun Chuanqi, Xie Guoxi, Verdonschot Nico, Sprengers André, Knee Bone and Cartilage Segmentation Based on a 3D Deep Neural Network Using Adversarial Loss for Prior Shape Constraint, Frontiers in Medicine, Vol. 9, DOI 10.3389/fmed.2022.792900, 2022.
Reviewer 2 Report
I would like to congratulate for an excellent study, this study will improve this field significantly. My recommendation is to accept this paper in current form.
Author Response
I would like to congratulate for an excellent study, this study will improve this field significantly. My recommendation is to accept this paper in current form.
Thank you very much! Nonetheless, we made a few changes that hopefully improve the paper.
Reviewer 3 Report
Authors should compare their results with many existing methods and should describe the differences and contributions of this article from the existing methods.
Author Response
Authors should compare their results with many existing methods and should describe the differences and contributions of this article from the existing methods.
Thank you for the comments. We have updated the paper with such a comparison and described better other methods.
Reviewer 4 Report
In this manuscript, Robert et al reported an algorithm to automatically segment bone templates from MRI instead of CT using a deep neural network architecture called 2.5D U-Net, which was interesting and presented multiple 3D mesh templates. However, the main limitation of this study is lack of the statistical analysis to justify the accuracy of this segment model.
The means and standard deviations of the absolute errors should be provided.
Author Response
In this manuscript, Robert et al reported an algorithm to automatically segment bone templates from MRI instead of CT using a deep neural network architecture called 2.5D U-Net, which was interesting and presented multiple 3D mesh templates. However, the main limitation of this study is lack of the statistical analysis to justify the accuracy of this segment model.
Statistical analysis was provided and compared to other algorithms.
The means and standard deviations of the absolute errors should be provided.
This information was provided in the new version.
Round 2
Reviewer 1 Report
The author revised this article carefully for addressing my concerns. I appreciate the efforts of the authors. Suggestion: there is still a slight problem, please double-check the paper before publication such as grammar.
Reviewer 3 Report
Question is solved.
Reviewer 4 Report
I am fine with this revision.